# Coumarin-Based Triapine Derivatives and Their Copper(II) Complexes: Synthesis, Cytotoxicity and mR2 RNR Inhibition Activity

**DOI:** 10.3390/biom11060862

**Published:** 2021-06-09

**Authors:** Iryna Stepanenko, Maria V. Babak, Gabriella Spengler, Marta Hammerstad, Ana Popovic-Bijelic, Sergiu Shova, Gabriel E. Büchel, Denisa Darvasiova, Peter Rapta, Vladimir B. Arion

**Affiliations:** 1Institute of Inorganic Chemistry, University of Vienna, Währinger Strasse 42, A-1090 Vienna, Austria; 2Drug Discovery Lab, Department of Chemistry, City University of Hong Kong, 83 Tat Chee Avenue, Hong Kong SAR 518057, China; mbabak@cityu.edu.hk; 3Department of Medical Microbiology, Albert Szent-Györgyi Health Center and Faculty of Medicine, University of Szeged, Semmelweis utca 6, H-6725 Szeged, Hungary; spengler.gabriella@med.u-szeged.hu; 4Section for Biochemistry and Molecular Biology, Department of Biosciences, University of Oslo, P.O. Box 1066, Blindern, NO-0316 Oslo, Norway; marta.hammerstad@ibv.uio.no; 5Faculty of Physical Chemistry, University of Belgrade, Studentski trg 12–16, 11158 Belgrade, Serbia; ana@ffh.bg.ac.rs; 6“Petru Poni” Institute of Macromolecular Chemistry, Aleea Gr. Ghica Voda 41A, 700487 Iasi, Romania; shova@icmpp.ro; 7ChemConsult GmbH, P.O. Box 43, 9485 Nendeln, Liechtenstein; gabriel.buechel@gmail.com; 8Faculty of Chemical and Food Technology, Institute of Physical Chemistry and Chemical Physics, Slovak University of Technology in Bratislava, Radlinského 9, SK-812 37 Bratislava, Slovakia; denisa.darvasiova@stuba.sk (D.D.); peter.rapta@stuba.sk (P.R.)

**Keywords:** triapine, coumarin, thiosemicarbazones, copper(II), electrochemistry, antiproliferative activity, tyrosyl radical reduction

## Abstract

A series of thiosemicarbazone-coumarin hybrids (**HL^1^-HL^3^** and **H_2_L^4^**) has been synthesised in 12 steps and used for the preparation of mono- and dinuclear copper(II) complexes, namely **Cu(HL^1^)Cl_2_** (**1**), **Cu(HL^2^)Cl_2_** (**2**), **Cu(HL^3^)Cl_2_** (**3**) and **Cu_2_(H_2_L^4^)Cl_4_** (**4**), isolated in hydrated or solvated forms. Both the organic hybrids and their copper(II) and dicopper(II) complexes were comprehensively characterised by analytical and spectroscopic techniques, i.e., elemental analysis, ESI mass spectrometry, 1D and 2D NMR, IR and UV–vis spectroscopies, cyclic voltammetry (CV) and spectroelectrochemistry (SEC). Re-crystallisation of **1** from methanol afforded single crystals of copper(II) complex with monoanionic ligand **Cu(L^1^)Cl**, which could be studied by single crystal X-ray diffraction (SC-XRD). The prepared copper(II) complexes and their metal-free ligands revealed antiproliferative activity against highly resistant cancer cell lines, including triple negative breast cancer cells MDA-MB-231, sensitive COLO-205 and multidrug resistant COLO-320 colorectal adenocarcinoma cell lines, as well as in healthy human lung fibroblasts MRC-5 and compared to those for triapine and doxorubicin. In addition, their ability to reduce the tyrosyl radical in mouse R2 protein of ribonucleotide reductase has been ascertained by EPR spectroscopy and the results were compared with those for triapine.

## 1. Introduction

Synthetic nucleoside analogues and Schiff bases are used as suitable models for investigation of nucleic acids and as chelating agents for application in different fields of research [1,2]. Schiff bases are easily prepared and their electronic and steric properties can be fine-tuned for biomedical applications. α-*N*-Heterocyclic thiosemicarbazones (TSCs) are excellent metal chelators, which act as mono- or polydentate ligands in metal complexes. They are well known for their broad spectrum of biological effects, including antitumour, antiviral, antifungal, antibacterial and antimalarial activity [3,4]. The most well-known representative of this class of compounds is 3-aminopyridine-2-carboxaldehyde thiosemicarbazone or triapine, (3-AP) ([Fig biomolecules-11-00862-ch001], left), which entered more than 30 phase I and II clinical trials [5]. 3-AP demonstrated excellent anticancer potential and was also shown to enhance the anticancer effects of other anticancer drugs, such as cisplatin, gemcitabine, doxorubicin, irinotecan, as well as the effect of radiotherapy [6,7,8,9,10]. The mechanism of action of 3-AP was linked to iron sequestration from the catalytic centre of ribonucleotide reductase (RNR) and transferrin, resulting in the formation of the inhibitory iron(II)-(3-AP)_2_ complex. It has been shown that this complex can generate reactive oxygen species (ROS), and that only catalytic amounts are needed for the complete reduction of the tyrosyl radical in mouse and human R2 RNR subunit [11,12,13,14,15]. TSCs were also shown to inhibit topoisomerase IIα (Topo IIα), which controls DNA topology upon cell division [16,17,18]. Quite recently, two other TSCs, namely di-2-pyridylketone 4-cyclohehyl-4-methyl-3-thiosemicarbazone (COTI-2) and (*E*)-*N’*-(6,7-dihydroquinolin-8(5*H*)-ylidene)-4-(pyridine-2-yl)piperazine-1-carbothiohydrazide (DpC), entered phase I clinical trials, reinforcing the interest to this class of compounds [19,20,21,22].

Despite the advances in preclinical development of TSCs, these compounds are still facing some fundamental problems, including low aqueous solubility and high toxicity [23,24,25]. The insightful application of high-impact molecular design elements might significantly shorten the optimisation process required to obtain highly efficacious drug candidates. For example, the insertion of a biologically active morpholine fragment into TSC backbones improved their aqueous solubility and anticancer activity in various cancer cell lines [23,26]. Herein, a biologically active coumarin (or 2*H*-chromen-2-one) fragment ([Fig biomolecules-11-00862-ch001], right) was selected to be incorporated, due to its well-documented wide spectrum of activities, including antibacterial, antifungal, neuroprotective, antiamoebic, anti-inflammatory, and cytotoxic properties [27,28,29,30,31,32,33,34,35,36,37,38]. In addition, another biologically active piperazine fragment was included, since heteroaromatic ring systems are considered highly impactful design elements for the optimisation of key pharmacological parameters [23,39].

A series of potentially bidentate (NS), tridentate (ONS or CNS, in particular, for Pd(II) and Ru(II)) Schiff bases is well documented in the literature [40,41,42,43,44,45,46,47,48,49,50]. These have been obtained by condensation reactions of acetyl-, formyl- or trifluoroacetyl-coumarins with substituted thiosemicarbazides. In the case of tridentate ONS Schiff bases, the starting coumarins contain a second carbonyl or hydroxyl function in a position suitable for chelation to a transition metal. Here, we propose a new synthetic pathway to coumarin-thiosemicarbazone hybrids with NNS binding site, as is the case for triapine, which is considered the most favourable for the design of potent metal-based anticancer drugs [11]. A series of novel antitumor TSC-coumarin hybrids and their Cu(II) complexes were prepared, by attachment of 3-AP-related moiety to a coumarin fragment at position 3 via a piperazine spacer. Copper(II) complexes with potentially tetradentate piperazine ligands bearing pendant pyridyl groups were reported to effectively cleave DNA and to be cytotoxic [51]. An additional 3-AP-based moiety was attached at position 7 (for both type of structures see [Fig biomolecules-11-00862-ch002]).

It is well known that these two frameworks separately exhibit anticancer activity, therefore the aim was to determine if they can work in synergy. It was hypothesised that the novel hybrid molecules could display multiple biological activities with an improved selectivity profile, and reduced side effects. The novel compounds were characterised by spectroscopic and spectroelectrochemical techniques, and their antiproliferative activity was screened by a colorimetric MTT assay in various cancer cell lines (human breast adenocarcinoma (MDA-MB-231), human colorectal adenocarcinoma cell lines (COLO-205 and COLO-320)) and human healthy lung fibroblasts (MRC-5). The inhibition of mouse R2 RNR protein, a likely target for some TSCs, by the metal-free ligand **HL^1^** and copper(II) complexes **1** and **3** (see [Fig biomolecules-11-00862-ch002]) was also investigated, and compared with that of triapine.

## 2. Experimental Part

### 2.1. Chemicals

2,4-Pyridinedicarboxylic acid, 2,4-dihydroxybenzaldehyde, diethyl malonate, *t*-butyl piperazine-1-carboxylate, 1-ethyl-3-(3-dimethylaminopropyl)carbodiimide (EDCI), 1-hydroxybenzotriazole (HOBt), diisopropylethylamine (DIEA), 4,4-dimethyl-3-thiosemicarbazide, 4-phenylthiosemicarbazide were purchased from Acros Organics (Fischer Scientific UK; Geel, Belgium), Alfa Aesar (Karlsruhe, Germany), Sigma-Aldrich (Schnelldorf, Germany) and/or Iris-Biotech (Marktredwitz, Germany). *N*-(4-Hydroxy-3,5-dimethylphenyl)hydrazinecarbothioamide was synthesised in five steps according to published protocols [52].

### 2.2. Synthesis of TSCs

The syntheses of 4-chloromethyl-2-dimethoxymethylpyridine (**E**) starting from 2,4-pyridinedicarboxylic acid in five steps (Appendix A) and 7-hydroxy-3-(piperazine-1-carbonyl)-2H-chromen-2-one (**H**) starting from 2,4-dihydroxybenzaldehyde in four steps (Appendix A) are given in details in ESI † (see also Appendix A).

*3-(4-((2-(Dimethoxymethyl)pyridin-4-yl)methyl)piperazine-1-carbonyl)-7-hydroxy-2H-chromen-2-one* (**I_1_** in Scheme 1). To a solution of 4-chloromethyl-2-dimethoxymethylpyridine **E** (0.92 g, 4.6 mmol) in a 1:1 mixture of dry CH_2_Cl_2_ and THF (50 mL), 7-hydroxy-3-(piperazine-1-carbonyl)-2*H*-chromen-2-one **H** (as **H**·TFA, 3.5 g, 9.01 mmol) and 1,1,3,3-tetramethylguanidine (TMG, 1.7 mL, 13.5 mmol) were added. The reaction mixture was stirred at 50 °C for 24 h. The solvent mixture was removed under reduced pressure and the brown oily residue was purified by column chromatography on silica by using CH_2_Cl_2_/MeOH 10:1 as eluent to give the side product **I_2_** (R_f_ = 0.64) as a cream-coloured solid (ca. 0.06 g) and **I_1_** (R_f_ = 0.63) as a bright-yellow solid (1.34 g, 67.3%). Other eluents can be also used: MeOH/EtOAc 2:1: fr_1_ is **I_1_** (R_f_ = 0.78), fr_2_ is a I_2_ (R_f_ = 0.67); acetone (used for TLC plates): fr_1_ is **I_1_** (R_f_ = 0.54), fr_2_ is **I_2_** (R_f_ = 0.34). Positive ion ESI-MS for **I_1_** (ACN/MeOH + 1% H_2_O): *m*/*z* 440.18 [M + H]^+^, 462.16 [M + Na]^+^; negative: *m*/*z* 437.98 [M–H]^−^. ^1^H NMR (**I_1_**, 600 MHz, DMSO-*d*_6_) δ, ppm: 8.48 (d, *J* = 5.5 Hz, 1H, H_19_), 8.05 (s, 1H, H_4_), 7.57 (d, *J* = 8.6 Hz, 1H, H_5_), 7.43 (s, 1H, H_21_), 7.33 (dd, *J* = 5.0, 1.5 Hz, 1H, H_18_), 6.81 (dd, *J* = 8.5, 2.2 Hz, 1H, H_6_), 6.73 (d, *J* = 2.1 Hz, 1H, H_8_), 5.27 (s, 1H, H_22_), 3.60 (s, 2H, H_12_ or H_15_), 3.57 (s, 2H, H_16_), 3.34 (s, 2H, H_12_ or H_15_ (overlapped by H_2_O signal, from ^1^H,^1^H-COSY)), 3.29 (s, 6H, H_23_ and H_23′_), 2.41 (s, 2H, H_13_ or H_14_), 2.35 (s, 2H, H_13_ or H_14_). ^13^C NMR (**I_1_**, 151 MHz, DMSO-*d*_6_) δ, ppm: 163.34 (C_2_ or C_11_), 162.41 (C_7_), 157.97 (C_2_ or C_11_), 157.04 (C_20_), 155.66 (C_9_), 148.76 (C_19_), 147.70 (C_17_), 143.04 (C_4_), 130.38 (C_5_), 123.71 (C_18_), 120.62 (C_21_), 119.59 (C_3_), 113.73 (C_6_), 110.63 (C_10_), 103.95 (C_22_), 102.03 (C_8_), 60.43 (C_16_), 53.50 (C_23_), 52.71 (C_13_ or C_14_, {2.35 ppm}), 52.15 (C_13_ or C_14_, {2.41 ppm}), 46.46 (C_12_ or C_15_, {3.34 ppm}), 41.32 (C_12_ or C_15_, {3.60 ppm}). For atom labelling used in the NMR resonances assignment of **I_1_**, see Appendix A, ESI †.

*7-((2-(Dimethoxymethyl)pyridin-4-yl)methoxy)-3-(4-((2-(dimethoxymethyl)pyridin-4-yl)methyl)piperazine-1-carbonyl)-2H-chromen-2-one* (**I_2_** in Scheme 1). To a solution of 4-chloromethyl-2-dimethoxymethylpyridine **E** (1.0 g, 4.9 mmol) in a 1:1 mixture of dry CH_2_Cl_2_/THF (50 mL), 7-hydroxy-3-(piperazine-1-carbonyl)-2H-chromen-2-one **H** (as **H**·TFA, 2.5 g, 6.4 mmol) and 1,1,3,3-tetramethylguanidine (TMG, 1.85 mL, 14.7 mmol) were added. The reaction mixture was stirred at 50 °C for 24 h. The solvents were removed under reduced pressure and the brown oily residue was purified on silica by using eluents specified previously for **I_1_**, to produce **I_2_** (0.27 g, 18.0%) and **I_1_** (1.14 g, 52.2%). Positive ion ESI-MS for I_2_ (ACN/MeOH + 1% H_2_O): *m*/*z* 605.28 [M + H]^+^, 627.28 [M + Na]^+^; negative: *m*/*z* 603.28 [M–H]^−^. ^1^H NMR (**I_2_**, 600 MHz, DMSO-*d*_6_) δ, ppm: 8.57 (d, *J* = 5.0 Hz, 1H, H_27_), 8.48 (d, *J* = 5.5 Hz, 1H, H_19_), 8.11 (s, 1H, H_4_), 7.70 (d, *J* = 8.7 Hz, 1H, H_5_), 7.56 (s, 1H, H_29_), 7.44 (dd, 1H, H_26_), 7.43 (s, 1H, H_21_), 7.33 (dd, *J* = 5.0, 1.5 Hz, 1H, H_18_), 7.13 (d, *J* = 2.4 Hz, 1H, H_8_), 7.10 (dd, *J* = 8.6, 2.4 Hz, 1H, H_6_), 5.36 (s, 2H, H_24_), 5.30 (s, 1H, H_30_), 5.27 (s, 1H, H_22_), 3.61 (s, 2H, H_12_ or H_15_), 3.58 (s, 2H, H_16_), 3.37 (s, 2H, H_12_ or H_15_), 3.31 (s, 6H, H_31_ and H_31′_), 3.29 (s, 6H, H_23_ and H_23′_), 2.42 (s, 2H, H_13_ or H_14_), 2.35 (s, 2H, H_13_ or H_14_). ^13^C NMR (**I_2_**, 151 MHz, DMSO-*d*_6_) δ, ppm: 163.04 (C_2_ or C_11_), 161.56 (C_7_), 157.76 (C_2_ or C_11_), 157.38 (C_28_), 157.04 (C_20_), 155.34 (C_9_), 149.03 (C_27_), 148.76 (C_19_), 147.69 (C_17_), 146.03 (C_25_), 142.50 (C_4_), 130.23 (C_5_), 123.71 (C_18_), 121.71 (C_26_), 121.31 (C_3_), 120.61 (C_21_), 118.67 (C_29_), 113.43 (C_6_), 112.32 (C_10_), 103.95 (C_22_), 103.89 (C_30_), 101.56 (C_8_), 68.22 (C_24_), 60.42 (C_16_), 53.54 (C_23_ or C_31_), 53.43 (C_23_ or C_31_), 52.70 (C_13_ or C_14_, {2.34 ppm}), 52.15 (C_13_ or C_14_, {2.42 ppm}), 46.40 (C_12_ or C_15_, {3.36 ppm}), 41.32 (C_12_ or C_15_, {3.60 ppm}). For atom labelling used in the NMR resonances assignment of **I_2_**, see Appendix A, ESI †.

*4-((4-(7-Hydroxy-2-oxo-2H-chromene-3-carbonyl)piperazin-1-yl)methyl)picolinaldehyde* (**J_1_** in Scheme 1), accompanied by *7-hydroxy-3-(4-((2-(hydroxy(methoxy)methyl)pyridin-4-yl)methyl)piperazine-1-carbonyl)-2H-chromen-2-one* (**J_1h_**). To a suspension of acetal **I_1_** (0.41 g, 0.93 mmol) in water (20 mL), 12 M HCl (0.23 mL, 2.76 mmol) was added. The yellow solution was heated at 60 °C for 3 h. Then, Et_3_N (0.4 mL, 2.87 mmol) was added. The solvent was removed under reduced pressure. The residue was purified on silica by using CH_2_Cl_2_/MeOH 10:1 as eluent. The first fraction as a mixture of hemiacetal **J_1h_** and aldehyde **J_1_** (R_f_ = 0.63) was collected as a cream-coloured solid. Yield: 0.29 g, 78.0%. The molar ratio of hemiacetal **J_1h_**/aldehyde **J_1_** is 1:6.5. Positive ion ESI-MS for **J_1h_** (ACN/MeOH + 1% H_2_O): *m*/*z* 394.13 [M + H]^+^, 416.12 [M + Na]^+^; negative: *m*/*z* 392.11 [M–H]^−^. ^1^H NMR (**J_1h_**, 600 MHz, DMSO-*d*_6_) δ, ppm: 10.75 (s, 1H, OH), 9.99 (s, 1H, H_22_), 8.76 (d, *J* = 5.0 Hz, 1H, H_19_), 8.07 (s, 1H, H_4_), 7.89 (s, 1H, H_21_), 7.67 (dd, *J* = 4.9, 1.5 Hz, 1H, H_18_), 7.58 (d, *J* = 8.6 Hz, 1H, H_5_), 6.82 (dd, *J* = 8.5, 2.3 Hz, 1H, H_6_), 6.74 (d, *J* = 2.2 Hz, 1H, H_8_), 3.67 (s, 2H, H_16_), 3.61 (s, 2H, H_12_ or H_15_), 3.37 (s, 2H, H_12_ or H_15_), 2.44 (s, 2H, H_13_ or H_14_), 2.39 (s, 2H, H_13_ or H_14_). ^13^C NMR (**J_1_**, 151 MHz, DMSO-*d*_6_) δ, ppm: 193.78 (C_22_), 163.27 (C_2_ or C_11_), 162.08 (C_7_), 157.93 (C_2_ or C_11_), 155.58 (C_9_), 152.48 (C_20_), 150.25 (C_19_), 149.07 (C_17_), 143.02 (C_4_), 130.39 (C_5_), 128.13 (C_18_), 121.35 (C_21_), 119.78 (C_3_), 113.60 (C_6_), 110.75 (C_10_), 102.01 (C_8_), 59.90 (C_16_), 52.73 (C_13_ or C_14_, {2.39 ppm}), 52.08 (C_13_ or C_14_, {2.44 ppm}), 46.45 (C_12_ or C_15_, {3.36 ppm}), 41.30 (C_12_ or C_15_, {3.61 ppm}). Positive ion ESI-MS for **J_1_** (ACN/MeOH + 1% H_2_O): *m*/*z* 426.16 [M + H]^+^, 448.14 [M + Na]^+^; negative: *m*/*z* 424.15 [M–H]^−^. ^1^H NMR (**J_1_**, 600 MHz, DMSO-*d*_6_) δ, ppm: 10.75 (s, 1H, OH), 8.43 (d, *J* = 5.6 Hz, 1H, H_19_), 8.07 (s, 1H, H_4_), 7.58 (d, *J* = 8.6 Hz, 1H, H_5_), 7.46 (s, 1H, H_21_), 7.28 (dd, *J* = 5.0, 1.5 Hz, 1H, H_18_), 6.82 (dd, *J* = 8.5, 2.3 Hz, 1H, H_6_), 6.74 (d, *J* = 2.2 Hz, 1H, H_8_), 6.71 (d, *J* = 7.3 Hz, 1H, OH (H_23_)), 5.38 (d, *J* = 7.2 Hz, 1H, H_22_), 3.61 (s, 2H, H_12_ or H_15_), 3.56 (s, 2H, H_16_), 3.37 (s, 2H, H_12_ or H_15_), 3.33 (s, 3H, H_24_), 2.43 (s, 2 H, H_13_ or H_14_), 2.36 (s, 2 H, H_13_ or H_14_). ^13^C NMR (**J_1_**, 151 MHz, DMSO-*d*_6_) δ, ppm: 163.28 (C_2_ or C_11_), 162.07 (C_7_), 159.78 (C_20_), 157.93 (C_2_ or C_11_), 155.59 (C_9_), 148.21 (C_19_), 147.61 (C_17_), 143.01 (C_4_), 130.39 (C_5_), 123.31 (C_18_), 119.98 (C_21_), 119.81 (C_3_), 113.61 (C_6_), 110.76 (C_10_), 102.02 (C_8_), 97.88 (C_22_), 60.55 (C_16_), 53.65 (C_24_), 52.73 (C_13_ or C_14_, {2.36 ppm}), 52.15 (C_13_ or C_14_, {2.43 ppm}), 46.45 (C_12_ or C_15_, {3.37 ppm}), 41.31 (C_12_ or C_15_, {3.61 ppm}). For atom labelling used in the NMR resonances assignment of **J_1_** and **J_1h_**, see Appendix A, ESI †. The mixture of hemiacetal **J_1h_** and aldehyde **J_1_** was successfully used in the next step (xii_1_).

*4-((4-(7-((2-Formylpyridin-4-yl)methoxy)-2-oxo-2H-chromene-3-carbonyl)piperazin-1-yl)methyl)picolinaldehyde* (**J_2_** in Scheme 1). To a suspension of diacetal **I_2_** (0.35 g, 0.58 mmol) in water (25 mL), 12 M HCl (0.25 mL, 3 mmol) was added. The yellow solution was heated at 60 °C for 4 h. Et_3_N (0.45 mL, 3.23 mmol) was added and then water was removed under reduced pressure. The residue was purified on silica by using a mixture of CH_2_Cl_2_/MeOH 8:1 as eluent. The first fraction with R_f_ ca. 0.73 as a mixture of **J_2_** (main product) and “hemiacetal and aldehyde” was collected as a cream-coloured solid. Yield: 0.163 g, 55.0% (calculated for **J_2_**). Positive ion ESI-MS for “hemiacetal-hemiacetal” (ACN/MeOH + 1% H_2_O): *m*/*z* 577.26 [M + H]^+^, 599.20 [M + Na]^+^. Positive ion ESI-MS for “hemiacetal-aldehyde” (ACN/MeOH + 1% H_2_O): *m*/*z* 545.21 [M + H]^+^, 567.19 [M + Na]^+^; negative: *m*/*z* 543.19 [M–H]^−^. Positive ion ESI-MS for “aldehyde-aldehyde”, **J_2_**: *m*/*z* 513.2 [M + H]^+^, 535.19 [M + Na]^+^; negative: *m*/*z* 511.18 [M–H]^−^. ^1^H NMR (**J_2_**, 600 MHz, DMSO-*d*_6_) δ, ppm: 10.01 (s, 1H, H_30_), 9.99 (s, 1H, H_22_), 8.84 (d, *J* = 4.9 Hz, 1H, H_27_), 8.77 (d, *J* = 4.8 Hz, 1H, H_19_), 8.12 (s, 1H, H_4_), 7.99 (s, 1H, H_29_), 7.89 (s, 1H, H_21_), 7.77 (dd, *J* = 5.0, 1.7 Hz, 1H, H_26_), 7.72 (d, *J* = 8.7 Hz, 1H, H_5_), 7.67 (dd, *J* = 4.9, 1.4 Hz, 1H, H_18_), 7.15 (d, *J* = 2.4 Hz, 1H, H_8_), 7.13 (dd, *J* = 8.6, 2.4 Hz, 1H, H_6_), 5.46 (s, 2H, H_24_), 3.67 (s, 2H, H_16_), 3.62 (s, 2H, H_12_ or H_15_), 3.39 (s, 2H, H_12_ or H_15_), 2.44 (s, 2H, H_13_ or H_14_), 2.39 (s, 2H, H_13_ or H_14_). ^13^C NMR (**J_2_**, 151 MHz, DMSO-*d*_6_) δ, ppm: 193.78 (C_22_), 193.56 (C_30_), 163.01 (C_2_ or C_11_), 161.36 (C_7_), 157.74 (C_2_ or C_11_), 155.33 (C_9_), 152.56 (C_20_), 152.48 (C_28_), 150.49 (C_27_), 150.26 (C_19_), 149.06 (C_17_), 147.23 (C_25_), 142.49 (C_4_), 130.28 (C_5_), 128.13 (C_18_), 126.08 (C_26_), 121.37 (C_3_), 121.35 (C_21_), 119.32 (C_29_), 113.37 (C_6_), 112.42 (C_10_), 101.60 (C_8_), 67.75 (C_24_), 59.89 (C_16_), 52.73 (C_13_ or C_14_, {2.39 ppm}), 52.08 (C_13_ or C_14_, {2.44 ppm}), 46.40 (C_12_ or C_15_, {3.39 ppm}), 41.32 (C_12_ or C_15_, {3.62 ppm}). Atom labelling used for the NMR resonances assignment of **J_2_** is shown in Appendix A, ESI †. The mixture of “hemiacetal and aldehyde” and aldehyde **J_2_** (Appendix A, ESI †) was successfully used for the next (xii_2_) step.

*2-((4-((4-(7-Hydroxy-2-oxo-2H-chromene-3-carbonyl)piperazin-1-yl)methyl)pyridin-2-yl)methylene)-N,N-dimethylhydrazine-1-carbothioamide* (**HL^1^·0.5H_2_O**). A suspension of aldehyde **J_1_** (260 mg, 0.66 mmol) and 4,4-dimethyl-3-thiosemicarbazide (79 mg, 0.66 mmol) in ethanol (10 mL) was heated at 80 °C for 2 h, then concentrated to ca. ½ of the original volume and stored overnight at 4 °C. The yellow product was filtered off and dried in vacuo (220 mg). The filtrate was evaporated and the residue was suspended in a small amount of water and an additional amount of the product was collected by filtration (39 mg). Total yield: 259 mg, 78.0%. Anal. Calcd for C_24_H_26_N_6_O_4_S·0.5H_2_O (*M_r_* = 503.57), %: C, 57.24; H, 5.40; N, 16.69; S, 6.37. Found, %: C, 57.01; H, 5.38; N, 16.76; S, 6.74. Positive ion ESI-MS (ACN/MeOH + 1% H_2_O): *m*/*z* 495.19 [M + H]^+^, 517.17 [M + Na]^+^, 533.14 [M + K]^+^; negative: *m*/*z* 493.16 [M–H]^−^. IR (ATR, selected bands, ν_max_, cm^−1^): 2924.98, 1715.91, 1604.23, 1571.96, 1460.70, 1314.99, 1220.21, 1143.61, 1001.04, 863.53. UV–vis (MeOH), λ_max_, nm (ε, M^−1^cm^−1^): 396 (2881), 331 (24112), 278 sh, 261 sh. ^1^H NMR (600 MHz, DMSO-*d*_6_, *Z*-isomer) δ, ppm: 15.12 (s, 1H, H_23_), 10.73 (s, 1H, OH), 8.71 (d, *J* = 5.1 Hz, 1H, H_19_), 8.07 (s, 1H, H_4_), 7.73 (s, 1H, H_21_), 7.63–7.55 (m, 2H, H_5_ + H_22_), 7.50 (d, *J* = 4.9 Hz, 1H, H_18_), 6.82 (dd, *J* = 8.5, 2.1 Hz, 1H, H_6_), 6.74 (d, *J* = 1.9 Hz, 1H, H_8_), 3.65 (s, 2H, H_16_), 3.61 (s, 2H, H_12_ or H_15_), 3.37 (s, 2H, H_12_ or H_15_), 3.36 (s, 6H, H_25_), 2.43 (s, 2H, H_13_ or H_14_), 2.39 (s, 2H, H_13_ or H_14_). ^13^C NMR (151 MHz, DMSO-*d*_6_, *Z*-isomer) δ, ppm: 180.08 (C_24_), 163.30 (C_2_ or C_11_), 162.08 (C_7_), 157.93 (C_2_ or C_11_), 155.59 (C_9_), 151.79 (C_20_), 149.98 (C_17_), 148.02 (C_19_), 143.05 (C_4_), 136.22 (C_22_), 130.39 (C_5_), 125.42 (C_21_), 123.99 (C_18_), 119.80 (C_3_), 113.60 (C_6_), 110.76 (C_10_), 102.01 (C_8_), 60.11 (C_16_), 52.76 (C_13_ or C_14_, {2.39 ppm}), 52.16 (C_13_ or C_14_, {2.44 ppm}), 46.44 (C_12_ or C_15_, {3.37 ppm}), 41.30 (C_12_ or C_15_, {3.61 ppm}), 40.69 (C_25_). ^1^H NMR (600 MHz, DMSO-*d*_6_, *E*-isomer) δ, ppm: 11.16 (s, 1H, H_23_), 10.73 (s, 1H, OH), 8.51 (d, *J* = 5.0 Hz, 1H, H_19_), 8.24 (s, 1H, H_22_), 8.07 (s, 1H, H_4_), 7.84 (s, 1H, H_21_), 7.63–7.55 (m, 1H, H_5_), 7.33 (d, *J* = 4.8 Hz, 1H, H_18_), 6.82 (dd, *J* = 8.5, 2.1 Hz, 1H, H_6_), 6.74 (d, *J* = 1.9 Hz, 1H, H_8_), 3.61 (s, 2H, H_12_ or H_15_), 3.59 (s, 2H, H_16_), 3.37 (s, 2H, H_12_ or H_15_), 3.32 (s, 6H, H_25_), 2.43 (s, 2H, H_13_ or H_14_), 2.39 (s, 2H, H_13_ or H_14_). ^13^C NMR (151 MHz, DMSO-*d*_6_, *E*-isomer) δ, ppm: 180.47 (C_24_), 163.27 (C_2_ or C_11_), 162.07 (C_7_), 157.93 (C_2_ or C_11_), 155.59 (C_9_), 153.66 (C_20_), 149.39 (C_19_), 147.72 (C_17_), 144.08 (C_22_), 143.01 (C_4_), 130.39 (C_5_), 123.95 (C_18_), 119.80 (C_3_), 119.09 (C_21_), 113.60 (C_6_), 110.76 (C_10_), 102.01 (C_8_), 60.35 (C_16_), 52.80 (C_13_ or C_14_, {2.39 ppm}), 52.16 (C_13_ or C_14_, {2.44 ppm}), 46.44 (C_12_ or C_15_, {3.37 ppm}), 42.16 (C_25_), 41.30 (C_12_ or C_15_, {3.61 ppm}). The molar ratio of *Z*-isomer/*E*-isomer in DMSO-*d*_6_ is 1:3.4. An overview of condensation reactions with thiosemicarbazides is shown in Appendix A, ESI †, while atom labelling used for the NMR resonances assignment is shown in Appendix A, ESI †. The line drawings for *Z*- and *E*-isomers of **HL^1^** are shown in [Fig biomolecules-11-00862-ch003].

*2-((4-((4-(7-Hydroxy-2-oxo-2H-chromene-3-carbonyl)piperazin-1-yl)methyl)pyridin-2-yl)methylene)-N-phenylhydrazine-1-carbothioamide* (**HL^2^·0.5C_2_H_5_OH·H_2_O**). A suspension of hemiacetal **J_1h_**/aldehyde **J_1_** as 1:4 mixture (237 mg, 0.6 mmol) and 4-phenylthiosemicarbazide (100.8 mg, 0.6 mmol) in ethanol (15 mL) was heated at 80 °C for 3 h and allowed to stand at 4 °C overnight. The yellow product was filtered off and dried in vacuo. Yield: 294 mg, 84%. Anal. Calcd for C_28_H_26_N_6_O_4_S·0.5C_2_H_5_OH·H_2_O (*M_r_* = 583.66), %: C, 59.68; H, 5.35; N, 14.39; S, 5.49. Found, %: C, 59.58; H, 5.34; N, 14.16; S, 5.38. Positive ion ESI-MS (ACN/MeOH + 1% H_2_O): *m*/*z* 543.19 [M + H]^+^, 565.18 [M + Na]^+^; negative: *m*/*z* 541.20 [M–H]^−^. IR (ATR, selected bands, ν_max_, cm^−1^): 3144.24, 1710.66, 1613.92, 1538.17, 1225.83, 1203.21, 1149.00, 1001.08, 926.26, 839.56. UV–vis (MeOH), λ_max_, nm (ε, M^−1^cm^−1^): 390 sh, 339 (38378), 260 sh. ^1^H NMR (600 MHz, DMSO-*d*_6_, *E*-isomer) δ, ppm: 12.01 (s, 1H, H_23_), 10.77 (s, 1H, OH), 10.20 (s, 1H, H_25_), 8.54 (d, *J* = 5.0 Hz, 1H, H_19_), 8.28 (s, 1H, H_21_), 8.20 (s, 1H, H_22_), 8.04 (s,1H, H_4_), 7.56 (d, *J* = 8.9 Hz, 1H, H_5_), 7.54 (d, *J* = 7.5 Hz, 2H, H_27_ + H_31_), 7.40 (m, 3H, H_28_ + H_30_ + H_18_), 7.24 (t, *J* = 7.4 Hz, 1H, H_29_), 6.80 (d, *J* = 8.4 Hz, 1H, H_6_), 6.71 (s, 1H, H_8_), 3.61 (s, 2H, H_12_ or H_15_), 3.59 (s, 2H, H_16_), 3.34 (s, 2H, H_12_ or H_15_), 2.43 (s, 2H, H_13_ or H_14_), 2.39 (s, 2H, H_13_ or H_14_). ^13^C NMR (151 MHz, DMSO-*d*_6_, *E*-isomer) δ, ppm: 176.48 (C_24_), 163.33 (C_2_ or C_11_), 162.82 (C_7_), 157.97 (C_2_ or C_11_), 155.68 (C_9_), 153.13 (C_20_), 149.37 (C_19_), 147.60 (C_17_), 143.20 (C_22_), 143.06 (C_4_), 138.95 (C_26_), 130.36 (C_5_), 128.16 (C_18_), 126.36 (C_27_ + C_31_), 125.67 (C_29_), 124.20 (C_28_ + C_30_), 120.53 (C_21_), 119.39 (C_3_), 113.78 (C_6_), 110.38 (C_10_), 102.03 (C_8_), 60.57 (C_16_), 52.82 (C_13_ or C_14_, {2.38 ppm}), 52.26 (C_13_ or C_14_, {2.43 ppm}), 46.43 (C_12_ or C_15_, {3.34 ppm}), 41.29 (C_12_ or C_15_, {3.61 ppm}). ^15^N NMR (61 MHz, DMSO-*d*_6_, *E*-isomer) δ, ppm: 174.27 (N_23_), 128.95 (N_25_). Atom labelling used for the NMR resonances assignment of **HL^2^** is shown in Appendix A, ESI †.

*7-Hydroxy-3-(4-((2-((2-(((4-hydroxy-3,5-dimethylphenyl)amino)methyl)hydrazineylidene)-methyl)pyridin-4-yl)methyl)piperazine-1-carbonyl)-2H-chromen-2-one* (**HL^3^·0.25C_2_H_5_OH·0.5H_2_O**). A suspension of hemiacetal **J_1h_**/aldehyde **J_1_** as 1:4 mixture (224 mg, 0.56 mmol) and N-(4-hydroxy-3,5-dimethylphenyl)hydrazinecarbothioamide (120.3 mg, 0.57 mmol) in ethanol (15 mL) was heated at 80 °C for 3 h, then concentrated to ca. 1/3 of original volume and allowed to stand at 4 °C overnight. The yellow precipitate was filtered off and dried in vacuo. Yield: 220 mg, 65.0%. Anal. Calcd for C_30_H_30_N_6_O_5_S·0.25C_2_H_5_OH·0.5H_2_O (*M_r_* = 607.19), %: C, 60.33; H, 5.39; N, 13.84; S, 5.28. Found, %: C, 60.17; H, 5.31; N, 14.03; S, 5.63. Positive ion ESI-MS (ACN/MeOH + 1% H_2_O): *m*/*z* 587.22 [M + H]^+^, 609.20 [M + Na]^+^; negative: *m*/*z* 585.20 [M–H]^−^. IR (ATR, selected bands, ν_max_, cm^−1^): 3280.19, 1695.04, 1615.80, 1542.25, 1484.85, 1468.79, 1221.47, 1197.21, 1004.82, 846.37. UV–vis (MeOH), λ_max_, nm (ε, M^−1^cm^−1^): 400 sh, 337 (36839), 260 sh. ^1^H NMR (600 MHz, DMSO-*d*_6_, *E*-isomer) δ, ppm: 11.85 (s, 1H, H_23_), 10.74 (s, 1H, OH), 9.94 (s, 1H, H_25_), 8.52 (d, *J* = 5.0 Hz, 1H, H_19_), 8.28 (s, 1H, H_21_), 8.23 (s, 1H, H_32_), 8.16 (s, 1H, H_22_), 8.05 (s, 1H, H_4_), 7.57 (d, *J* = 8.6 Hz, 1H, H_5_), 7.38 (d, *J* = 6.0 Hz, 1H, H_18_), 6.99 (s, 2H, H_27_ + H_31_), 6.82 (dd, *J* = 8.5, 2.3 Hz, 1H, H_6_), 6.74 (d, *J* = 2.2 Hz, 1H, H_8_), 3.60 (s, 2H, H_12_ or H_15_), 3.58 (s, 2H, H_16_), 3.36 (s, 2H, H_12_ or H_15_), 2.43 (s, 2H, H_13_ or H_14_), 2.37 (s, 2H, H_13_ or H_14_), 2.17 (s, 6H, CH_3_). ^13^C NMR (151 MHz, DMSO-*d*_6_, *E*-isomer) δ, ppm: 176.71 (C_24_), 163.26 (C_2_ or C_11_), 162.08 (C_7_), 157.93 (C_2_ or C_11_), 155.58 (C_9_), 153.27 (C_20_), 151.23 (C_29_), 149.30 (C_19_), 147.53 (C_17_), 143.00 (C_4_), 142.62 (C_22_), 130.39 (C_5_), 130.15 (C_26_), 126.60 (C_27_ + C_31_), 124.06 (C_18_), 123.89 (C_28_ + C_30_), 120.50 (C_21_), 119.79 (C_3_), 113.61 (C_6_), 110.74 (C_10_), 102.01 (C_8_), 60.56 (C_16_), 52.80 (C_13_ or C_14_, {2.37 ppm}), 52.26 (C_13_ or C_14_, {2.43 ppm}), 46.42 (C_12_ or C_15_, {3.35 ppm}), 41.29 (C_12_ or C_15_, {3.60 ppm}), 16.59 (C_CH3_). ^15^N NMR (61 MHz, DMSO-*d*_6_, *E*-isomer) δ, ppm: 173.22 (N_23_), 127.65 (N_25_). Atom labelling used for the NMR resonances assignment of **HL^3^** is shown in Appendix A, ESI †.

*2-((4-((4-(7-((2-((2-(dimethylcarbamothioyl)hydrazineylidene)methyl)pyridin-4-yl)methoxy)-2-oxo-2H-chromene-3-carbonyl)piperazin-1-yl)methyl)pyridin-2-yl)methylene)-N,N-dimethylhydrazine-1-carbothioamide* (**H_2_L^4^·0.5C_2_H_5_OH·0.75H_2_O**). A suspension of dialdehyde **J_2_** (150 mg, 0.29 mmol) and 4,4-dimethyl-3-thiosemicarbazide (70 mg, 0.59 mmol) in ethanol (15 mL) was heated at 80 °C for 2 h. The solvent was removed under reduced pressure and the yellow residue was suspended in water (5 mL), filtered off and dried in vacuo. Yield: 150 mg, 69.0%. Anal. Calcd for C_34_H_38_N_10_O_4_S_2_·0.5C_2_H_5_OH·0.75H_2_O (*M_r_* = 751.41), %: C, 55.94; H, 5.70; N, 18.64; S, 8.53. Found, %: C, 56.05; H, 5.37; N, 18.31; S, 8.68. Positive ion ESI-MS (ACN/MeOH + 1% H_2_O): *m*/*z* 715.25 [M + H]^+^, 737.24 [M + Na]^+^; negative: *m*/*z* 713.26 [M–H]^−^. IR (ATR, selected bands, ν_max_, cm^−1^): 2924.29, 1718.17, 1605.17, 1363.66, 1311.27, 1287.79, 1220.09, 1146.74, 1003.07, 864.65. UV–vis (MeOH), λ_max_, nm (ε, M^−1^cm^−1^): 330 (15870), 270 sh. ^1^H NMR (600 MHz, DMSO-*d*_6_, Z,Z-isomer) δ, ppm: 15.12 (s, 1H, H_31_), 15.06 (s, 1H, H_34_), 8.79 (d, *J* = 5.1 Hz, 1H, H_27_), 8.71 (d, *J* = 5.0 Hz, 1H, H_19_), 8.13 (s, 1H, H_4_), 7.84 (s, 1H, H_29_), 7.75–7.68 (m, 2H, H_5_ + H_21_), 7.63 (s, 1H, H_30_), 7.60 (s, 1H, H_22_), 7.59 (dd, *J* = 5.1, 1.4 Hz, 1H, H_26_), 7.50 (dd, *J* = 5.2, 1.1 Hz, 1H, H_18_), 7.15 (d, *J* = 2.1 Hz, 1H, H_8_), 7.11 (dd, *J* = 8.7, 2.4 Hz, 1H, H_6_), 5.43 (s, 2H, H_24_), 3.65 (s, 2H, H_16_), 3.62 (s, 2H, H_12_ or H_15_), 3.38 (s, 2H, H_12_ or H_15_), 3.39 (s, 6H, H_33_ or H_36_), 3.36 (s, 6H, H_33_ or H_36_), 2.44 (s, 2H, H_13_ or H_14_), 2.39 (s, 2H, H_13_ or H_14_). ^13^C NMR (151 MHz, DMSO-*d*_6_, *Z*,*Z*-isomer) δ, ppm: 180.08 (C_35_ or C_32_), 180.07 (C_35_ or C_32_), 163.05 (C_2_ or C_11_), 161.53 (C_7_), 157.75 (C_2_ or C_11_), 155.32 (C_9_), 151.89 (C_20_ or C_28_), 151.79 (C_20_ or C_28_), 149.98 (C_17_), 148.32 (C_27_), 148.03 (C_19_), 147.69 (C_25_), 142.55 (C_4_), 136.22 (C_22_), 135.88 (C_30_), 130.25 (C_5_), 125.42 (C_21_), 123.99 (C_18_), 123.38 (C_29_), 121.92 (C_26_), 121.30 (C_3_), 113.41 (C_6_), 112.34 (C_10_), 101.66 (C_8_), 67.91 (C_24_), 60.11 (C_16_), 52.80 (C_13_ or C_14_, {2.39 ppm}), 52.16 (C_13_ or C_14_, {2.44 ppm}), 46.40 (C_12_ or C_15_, {3.38 ppm}), 41.31 (C_12_ or C_15_, {3.62 ppm}), 40.86 (C_30_ + C_36_). ^1^H NMR (600 MHz, DMSO-*d*_6_, *E*,*E*-isomer) δ, ppm: 11.19 (s, 1H, H_34_), 11.16 (s, 1H, H_31_), 8.59 (d, *J* = 5.1 Hz, 1H, H_27_), 8.51 (d, *J* = 5.0 Hz, 1H, H_19_), 8.25 (s, 1H, H_30_), 8.24 (s, 1H, H_22_), 8.12 (s, 1H, H_4_), 7.94 (s, 1H, H_29_), 7.84 (s, 1H, H_21_), 7.75–7.68 (m, 1H, H_5_), 7.43 (dd, *J* = 5.1, 1.4 Hz, 1H, H_26_), 7.33 (dd, *J* = 5.0, 1.2 Hz, 1H, H_18_), 7.15 (d, *J* = 2.1 Hz, 1H, H_8_), 7.11 (dd, *J* = 8.7, 2.4 Hz, 1H, H_6_), 5.39 (s, 2H, H_24_), 3.62 (s, 2H, H_12_ or H_15_), 3.59 (s, 2H, H_16_), 3.38 (s, 2H, H_12_ or H_15_), 3.31 (s, 6H, H_33_ or H_36_), 3.30 (s, 6H, H_33_ or H_36_), 2.44 (s, 2H, H_13_ or H_14_), 2.39 (s, 2H, H_13_ or H_14_). ^13^C NMR (151 MHz, DMSO-*d*_6_, *E*,*E*-isomer) δ, ppm: 180.51 (C_35_), 180.47 (C_32_), 163.03 (C_2_ or C_11_), 161.53 (C_7_), 157.75 (C_2_ or C_11_), 155.32 (C_9_), 153.86 (C_20_ or C_28_), 153.67 (C_20_ or C_28_), 149.68 (C_27_), 149.40 (C_19_), 147.72 (C_17_), 145.97 (C_25_), 144.09 (C_22_ or C_30_), 143.67 (C_22_ or C_30_), 142.51 (C_4_), 130.25 (C_5_), 123.95 (C_18_), 121.84 (C_26_), 121.30 (C_3_), 119.10 (C_21_), 117.17 (C_29_), 113.41 (C_6_), 112.34 (C_10_), 101.60 (C_8_), 68.11 (C_24_), 60.34 (C_16_), 52.80 (C_13_ or C_14_, {2.39 ppm}), 52.16 (C_13_ or C_14_, {2.44 ppm}), 46.40 (C_12_ or C_15_, {3.38 ppm}), 42.18 (C_30_ + C_36_), 41.31 (C_12_ or C_15_, {3.62 ppm}). ^15^N NMR (61 MHz, DMSO-*d*_6_, *E*,*E*-isomer) δ, ppm: 172.37 (N_31_ + N_34_). The molar ratio of *Z*,*Z*-isomer/*E*,*E*-isomer of **H_2_L^4^** in DMSO-*d*_6_ is 1:4.3. Atom labelling used for the NMR resonances assignment of **H_2_L^4^** is shown in Appendix A, ESI †.

### 2.3. Synthesis of the Copper(II) Complexes

**Cu(HL^1^)Cl_2_·2.25H_2_O (1·2.25H_2_O).** A solution of CuCl_2_·2H_2_O (34.5 mg, 0.2 mmol) in methanol (2 mL) was added to a warm solution (40–50 °C) of **HL^1^** (100 mg, 0.2 mmol) in methanol (20 mL). The resulting suspension was stirred at room temperature for 3 h and then left to stand at 4 °C overnight. The greenish precipitate of 1 was filtered off, washed with a small amount of methanol and dried in vacuo at 40 °C. Yield: 100 mg, 75.0%. Anal. Calcd for C_24_H_26_Cl_2_CuN_6_O_4_S·2.25H_2_O (*M_r_* = 669.55), %: C, 43.05; H, 4.59; N, 12.55; S, 4.79. Found, %: C, 42.95; H, 4.36; N, 12.49; S, 5.03. Positive ion ESI-MS (ACN/MeOH + 1% H_2_O): *m*/*z* 556.13 [Cu(HL^1^)]^+^; negative: *m*/*z* 590.12 [Cu(L^1^)Cl–H]^−^. IR (ATR, selected bands, *ν*_max_, cm^−1^): 3335.23, 1707.99, 1606.25, 1570.08, 1510.03, 1366.67, 1223.68, 1120.81, 958.56, 914.56. UV–vis (MeOH), λ_max_, nm (ε, M^−1^cm^−1^): 420 (16097), 342 (29742), 315 sh, 249 sh. The light brown single crystals of **Cu(L^1^)Cl·1.775H_2_O (1′·1.775H_2_O)** were obtained by re-crystallisation of **1·2.25H_2_O** from methanol.

**Cu(HL^2^)Cl_2_·2.25H_2_O (2·2.25H_2_O).** A solution of CuCl_2_·2H_2_O (31.4 mg, 0.18 mmol) in ethanol (2 mL) was added to a warm solution of **HL^2^** (100 mg, 0.18 mmol) in ethanol (15 mL). (**HL^2^** was originally dissolved in ethanol (80 mL) at 70 °C, then this solution was concentrated to ca. 15 mL at 40 °C.) The resulting suspension was stirred at room temperature for 2 h and allowed to stand overnight at 4 °C. The brown precipitate was filtered off, washed with ethanol (2 mL) and dried in vacuo. Yield: 100 mg, 77.0%. Anal. Calcd for C_28_H_26_Cl_2_CuN_6_O_4_S·2.25H_2_O (*M_r_* = 717.59), %: C, 46.86; H, 4.28; N, 11.71; S, 4.47; Cl, 9.88. Found, %: C, 46.54; H, 3.91; N, 11.41; S, 4.57; Cl, 9.71. Positive ion ESI-MS (ACN/MeOH + 1% H_2_O): *m*/*z* 604.12 [Cu(L^2^)]^+^; negative: *m*/*z* 638.11 [Cu(L^2^)Cl–H]^−^. IR (ATR, selected bands, *ν*_max_, cm^−1^): 3043.82, 1708.93, 1603.13, 1501.19, 1419.14, 1343.40, 1308.50, 1220.69, 1120.12, 958.32. UV–vis (MeOH), λ_max_, nm (ε, M^−1^cm^−1^): 415 (19464), 345 (32857), 250 sh.

**Cu(HL^3^)Cl_2_·0.5C_2_H_5_OH·2.5H_2_O (3·0.5C_2_H_5_OH·2.5H_2_O).** A solution of CuCl_2_·2H_2_O (29.1 mg, 0.17 mmol) in ethanol (2 mL) was added to a solution of **HL^3^** (100 mg, 0.17 mmol) in warm ethanol (20 mL). (**HL^3^** was first dissolved in ethanol (75 mL) at 70 °C, then this solution was concentrated to ca. 20 mL at 40 °C.) The resulting brown solution was stirred at room temperature overnight, then concentrated to ½ of original volume and allowed to stand at 4 °C for 72 h. The brown precipitate was filtered off, washed with ethanol (2 mL) and dried in vacuo. Yield: 110 mg, 82.0%. Anal. Calcd for C_30_H_30_Cl_2_CuN_6_O_5_S·0.5C_2_H_5_OH·2.5H_2_O (*M_r_* = 789.19), %: C, 47.18; H, 4.85; N, 10.65; S, 4.06. Found, %: C, 46.89; H, 4.45; N, 10.29; S, 4.17. Positive ion ESI-MS (ACN/MeOH + 1% H_2_O): *m*/*z* 648.16 [Cu(L^3^)]^+^; negative: *m*/*z* 682.13 [Cu(L^3^)Cl–H]^−^. IR (ATR, selected bands, *ν*_max_, cm^−1^): 3287.21, 2968.62, 1704.98, 1605.13, 1568.45, 1416.15, 1219.17, 1119.15, 1040.61, 849.77. UV–vis (MeOH), λ_max_, nm (ε, M^−1^cm^−1^): 422 (12125), 343 (18473), 288 (13458), 258 sh.

**Cu_2_(H_2_L^4^)Cl_4_·1.5CH_3_OH·2.5H_2_O (4·1.5CH_3_OH·2.5H_2_O).** CuCl_2_·2H_2_O (28.6 mg, 0.17 mmol) was added to a suspension of **H_2_L^4^** (60 mg, 0.08 mmol) in DMF (15 mL) at 50 °C. The resulting brown solution was stirred at room temperature for 2 h and the solvent was removed under reduced pressure. The green-isch residue was suspended in a small amount of methanol (5 mL), filtered off and dried in vacuo. Yield: 68 mg, 79.0%. Anal. Calcd for C_34_H_38_Cl_4_Cu_2_N_10_O_4_S_2_·1.5CH_3_OH·2.5H_2_O (*M_r_* = 1076.87), %: C, 39.59; H, 4.59; N, 13.01; S, 5.96. Found, %: C, 39.89; H, 4.52; N, 13.05; S, 5.69. Positive ion ESI-MS (ACN/MeOH + 1% H_2_O): *m*/*z* 419.02 [Cu_2_(L^4^)]^2+^, 875.06 [Cu_2_(L^4^)Cl]^+^; negative: *m*/*z* 944.96 [Cu_2_(L^4^)Cl_3_]^−^. IR (ATR, selected bands, *ν*_max_, cm^−1^): 3519.95, 3451.46, 3030.62, 2927.48, 1704.81, 1606.91, 1506.32, 1371.97, 1243.96, 1126.58, 1046.47, 933.37, 850.79. UV–vis (MeOH), λ_max_, nm (ε, M^−1^cm^−1^): 426 (19893), 340 sh, 318 (25679), 254 sh.

### 2.4. Physical Measurements

Elemental analyses for **1–4** were carried out in a Carlo Erba microanalyser at the Microanalytical Laboratory of the University of Vienna. TFA content was measured by using free zone capillary electrophoresis (Agilent Scientific Instruments, Santa Clara, CA 95051,United States). Separation of ions was achieved at −30 kV along a 45 cm silica capillary with 50 µm bore width. The electrolyte was a Good’s buffer solution prepared from 50 mM cyclohexylaminosulfonic acid and 20 mM arginine (pH = 9.1). Trimethyl(tetradecyl)ammonium hydroxide was added as EOF modifier also influencing the separation of TFA from acetate. Electrospray ionisation mass spectrometry (ESI-MS) was carried out with an amaZon speed ETD Bruker (Bruker Daltonik GmbH, Bremen, Germany, *m*/*z* range 0–900, ion positive/negative mode, 180 °C, heating gas N2 (5 L/min), Capillary 4500 V, End Plate Offset 500 V) instrument for all compounds. Expected and experimental isotopic distributions were compared. UV–vis spectra of **HL^1^-HL^3^** and **H_2_L^4^**, **1–4** were measured on a Perkin Elmer UV–Vis spectrophotometer Lambda 35 in the 240 to 700 nm window using samples dissolved in methanol. IR spectra of **1–4** were recorded on a Bucker Vertex 70 Fourier transform IR spectrometer (4000–600 cm^−1^) using the ATR technique. 1D (^1^H, ^13^C) and 2D (^1^H-^1^H COSY, ^1^H-^1^H TOCSY, ^1^H-^1^H NOESY, ^1^H-^13^C HSQC, ^1^H-^13^C HMBC, ^1^H-^15^N HSQC, ^1^H-^15^N HMBC) NMR spectra of intermediates and **HL^1^-HL^3^** and **H_2_L^4^** were measured on a Bruker AV (Bruker BioSpin GmbH, Rheinstetten, Germany) NEO 500 or AV III 600 spectrometers in DMSO-*d*_6_ at 25 °C. Fluorescence excitation and emission spectra of **HL^1^-HL^3^** and **H_2_L^4^** were recorded in H_2_O, 1% DMSO/H_2_O solutions with a Horiba FluoroMax-4 (HORIBA Jobin Yvon GmbH, Unterhaching, Germany) spectrofluorimeter and processed using the FluorEssence v3.5 software package.

### 2.5. Crystallographic Structure Determination

X-ray diffraction measurement of **Cu(L^1^)Cl·1.775H_2_O (1′·1.775H_2_O)** was performed on a Bruker D8 (Karlsruhe, Germany) Venture diffractometer. A single crystal was positioned at 45 mm from the detector, and 1591 frames were measured, each for 60 s over 0.7° scan width. Crystal data, data collection parameters, and structure refinement details are given in Table 1. The structure was solved by direct methods and refined by full-matrix least-squares techniques. Non-H atoms were refined with anisotropic displacement parameters except those belonging to disordered fragments. H atoms were inserted in calculated positions and refined with a riding model. The following computer programs and hardware were used: structure solution, *SHELXS-2014* and refinement, *SHELXL-2014* [53] molecular diagrams, ORTEP [54] computer, Intel CoreDuo. CCDC 2047825.

### 2.6. Electrochemistry and Spectroelectrochemistry

Cyclic voltammetry experiments with 0.5 mM solutions of Cu(II) complexes and ligands in 0.1 M *n*Bu_4_NPF_6_ DMSO were performed under argon atmosphere using a three-electrode arrangement with glassy carbon 1 mm disc working electrode (from Ionode, Australia), platinum wire as counter electrode, and silver wire as pseudo-reference electrode. Ferrocene (Fc) served as the internal potential standard. A Heka PG310USB (Lambrecht, Germany) potentiostat with a PotMaster 2.73 software package served as the potential control in voltammetry studies. In situ ultraviolet-visible-near-infrared (UV‒vis‒NIR) spectroelectrochemical measurements were performed on a spectrometer Avantes (Apeldoorn, The Netherlands) (Model AvaSpec-2048x14-USB2) in the spectroelectrochemical cell kit (AKSTCKIT3) with the Pt-microstructured honeycomb working electrode (1.7 mm optical path length), purchased from Pine Research Instrumentation (Lyon, France). The cell was positioned in the CUV‒UV Cuvette Holder (Ocean Optics, Ostfildern, Germany) connected to the diode-array UV‒vis‒NIR spectrometer by optical fibres. UV‒vis‒NIR spectra were processed using the AvaSoft 7.7 software package. Halogen and deuterium lamps were used as light sources (Avantes, Model AvaLight-DH-S-BAL). The in situ EPR spectroelectrochemical experiments were carried out under an argon atmosphere in the EPR flat cell (0.5 mm thickness of the inner space) equipped with a large platinum mesh working electrode. The freshly prepared solutions were carefully purged with argon and the electrolytic cell was polarised in the galvanostatic mode directly in the cylindrical EPR cavity TM-110 (ER 4103 TM) and the electron paramagnetic resonance (EPR) spectra were measured in situ. The EPR spectra were recorded at room temperature or at 77 K with the EMX Bruker spectrometer (Rheinstetten, Germany).

### 2.7. Cell Lines and Culture Conditions

Human breast adenocarcinoma cells MDA-MB-231, sensitive COLO-205 and multidrug resistant COLO-320 colorectal adenocarcinoma cells and normal human lung fibroblasts MRC-5 were obtained from ATCC. COLO-205 and COLO-320 cells were cultured in RPMI-1640 medium containing 10% foetal bovine serum, MRC-5 cells were cultured in EMEM medium containing 10% foetal bovine serum. MDA-MB-231 cells were cultured in DMEM/high glucose medium containing 10% FBS. Adherent MDA-MB-231 cells were grown in Falcon tissue culture 75 cm^2^ flasks and all other cells were grown in tissue culture 25 cm^2^ flasks (BD Biosciences, Singapore). All cell lines were grown at 37 °C in a humidified atmosphere of 95% air and 5% CO_2_. The stock solutions of copper(II) complexes were prepared in DMSO.

### 2.8. Inhibition of Cell Viability Assay

The cytotoxicity of the compounds was determined by colorimetric MTT assay. The cells were harvested from culture flasks by trypsinisation and seeded into Cellstar 96-well microculture plates at the seeding density of 6000 cells per well (6 × 10^4^ cells/mL, MDA-MB-231) or 10,000 cells per well (1× 10^5^ cells/mL cells/mL, COLO-205, COLO-320 and MRC-5). The cells were allowed to resume exponential growth for 24 h, and subsequently were exposed to drugs at different concentrations in media for 72 h. The drugs were diluted in complete medium at the desired concentration and added to each well (100 µL) and serially diluted to other wells. After exposure for 72 h, the media were replaced with MTT in media (5 mg/mL, 100 µL) and incubated for additional 50 min. Subsequently, the media were aspirated and the purple formazan crystals formed in viable cells were dissolved in DMSO (100 µL). Optical densities were measured at 570 nm using the BioTek H1 Synergy (BioTek, Singapore) microplate reader. The quantity of viable cells was expressed in terms of treated/control (T/C) values in comparison to untreated control cells, and 50% inhibitory concentrations (IC_50_) were calculated from concentration-effect curves by interpolation. Evaluation was based on at least three independent experiments, each comprising six replicates per concentration level.

### 2.9. Tyrosyl Radical Reduction in Mouse R2 RNR Protein

The 9.4 GHz EPR spectra were recorded at 30 K on a Bruker Elexsys II E540 EPR spectrometer with an Oxford Instruments ER 4112HV helium cryostat, essentially as described previously [13]. The concentration of the tyrosyl radical in mouse R2 ribonucleotide reductase protein (mR2) was determined by double integration of EPR spectra recorded at non-saturating microwave power levels (3.2 mW) and compared with the copper standard [55] mR2 protein was expressed, purified, and iron-reconstituted as described previously [56] and passed through a 5 mL HiTrap desalting column (GE Healthcare) to remove excess iron. The purified, iron-reconstituted mR2 protein resulted in the formation of 0.76 tyrosyl radical/polypeptide. Samples containing 20 µM mR2 in 50 mM Hepes buffer, pH 7.50/100 mM NaCl, and 20 µM **HL^1^**, complex **1**, or complex **3** in 1% (*v*/*v*) DMSO/H_2_O, and 2 mM dithiothreitol (DTT) were incubated for indicated times and quickly frozen in cold isopentane. The same samples were used for repeated incubations at room temperature. The experiments were performed in duplicates.

### 2.10. Redox Activity of **[Fe^II^(L^1^)_2_]**

The generation of paramagnetic intermediates was monitored by cw-EPR spectroscopy using the EMX spectrometer (Bruker). Deionised water was used for the preparation of ethanol/water solutions in which **HL^1^** was dissolved (ethanol was used to increase the solubility of **HL^1^**). The spin trapping agent (DMPO; Sigma-Aldrich) was distilled prior to use.

## 3. Results and Discussion

### 3.1. Synthesis and Characterisation of **HL^1^-HL^3^** and **H_2_L^4^**, Their Copper(II) Complexes **1–4** and **1**′

The synthesis of ligands **HL^1^-HL^3^** and **H_2_L^4^** was realised in 12 steps. Two building blocks were prepared first, namely 4-chloromethyl-2-dimethoxymethylpyridine (**E**) starting from 2,4-pyridinedicarboxylic acid in five steps, as shown in Appendix A, and 7-hydroxy-3-(piperazine-1-carbonyl)-2H-chromen-2-one (**H**) starting from 2,4-dihydroxybenzaldehyde in four steps, as shown in Appendix A and discussed in detail in ESI †. The atom labelling of the precursors for NMR resonances assignment is given in Appendix A. Then these building blocks **E** and **H** were combined to give aldehydes or aldehyde precursors, which entered condensation reactions with thiosemicarbazide derivatives with formation of **HL^1^-HL^3^** and **H_2_L^4^** in 3 steps as shown in Scheme 1.

#### 3.1.1. Synthesis of 3-(4-((2-(Dimethoxymethyl)pyridin-4-yl)methyl)piperazine-1-carbonyl)-7-hydroxy-2H-chromen-2-one (**I_1_**) and 7-((2-(Dimethoxymethyl)pyridin-4-yl)methoxy)-3-(4-((2-(dimethoxymethyl)pyridin-4-yl)methyl)piperazine-1-carbonyl)-2H-chromen-2-one (**I_2_**)

The condensation of two building blocks **H** and **E** (Scheme 1) was first attempted in the presence of Et_3_N (pK_a_ = 10.75) to yield the desired product **I_1_** in low yield (12–17%). By using a much stronger base, namely 1,1,3,3-tetramethylguanidine (TMG, pK_a_ = 13), and optimising the molar ratio of reactants the **I_1_** was produced in good to very good yield (50–80%) (Appendix A, ESI †). We also noticed that the synthesis of **I_1_** was accompanied by the formation of another product of condensation, i.e., **I_2_**, with two pyridine rings and one coumarin-piperazine moiety even if a double excess of **H** in comparison to **E** was used. Interestingly, the same reaction performed in the presence of K_2_CO_3_ as a base resulted only in **I_2_**. The two products **I_1_** and **I_2_** were separated by column chromatography with MeOH/EtOAc 2:1 as eluent (for details see Experimental part). The identity of **I_1_** and **I_2_** was confirmed by positive ESI mass spectra, which showed peaks at *m*/*z* 440.18, 462.16 for **I_1_** and 605.28, 627.28 for **I_2_**, attributed to [M + H]^+^and [M + Na]^+^, respectively. ^1^H and ^13^C NMR spectra were also in agreement with the structures proposed. Assignment of proton resonances was carried out based on 2D NMR experiments. The presence of the second pyridine ring-containing fragment instead of the hydroxyl group at the position 7 of the coumarin ring in **I_2_** was evidenced by an additional group of signals due to 2-dimethoxymethylpyridine moiety (Atom_24_-Atom_31_, Appendix A, ESI †), as well as by the downfield shifts of the nearest to the condensation place protons H_6_ and H_8_ at 6.81, 6.73 ppm in **I_1_** and at 7.10 and 7.13 ppm in **I_2_**. The rest of protons (Atom_4_-Atom_23_) of **I_2_** revealed the same resonances in terms of chemical shifts as the corresponding protons in **I_1_**. Even the methylene protons H_16_ in **I_1_** and **I_2_** resonate at almost the same magnetic field, namely at 3.57 and 3.58 ppm, respectively. The methylene protons H_24_ in **I_2_** are seen at 5.36 ppm and can be used as diagnostic signature for identification of **I_2_**. The corresponding methylene carbon atoms in **I_2_** also showed noticeable differences in resonances with C_16_ at 60.43 ppm and C_24_ at 68.22 ppm. The downfield shift of C_24_ resonance is in accordance with the vicinity of a more electronegative oxygen atom next to Atom_24_ in comparison to nitrogen atom next to Atom_16_ (Appendix A, ESI †).

#### 3.1.2. Synthesis of 4-((4-(7-Hydroxy-2-oxo-2H-chromene-3-carbonyl)piperazin-1-yl)methyl)picolinaldehyde (**J_1_**), Accompanied by Generation of 7-Hydroxy-3-(4-((2-(hydroxy(methoxy)methyl)pyridin-4-yl)methyl)piperazine-1-carbonyl)-2H-chromen-2-one (**J_1h_**) 

Several approaches for deacetalisation of acetal **I_1_** were explored (Scheme 1). First, the cleavage of acetals was attempted in water at 90–100 °C for 24 h or in a water-acetone mixture in the presence of amberlite [57,58,59] a well-known acid resin catalyst, at room temperature for 3 h. However, under these conditions the acetal **I_1_** remained intact. When, however, hydrolysis of **I_1_** was performed in water, in the presence of 12 M HCl (molar ratio of **I_1_**/HCl 1:3, 60 °C, 3 h) an equilibrium between the hemiacetal **J_1h_** and the aldehyde **J_1_** was reached. The progress of hydrolysis was monitored by ESI MS (**J_1h_**: positive ion peaks at *m*/*z* 426.16 [M + H]^+^, 448.14 [M + Na]^+^; **J_1_**: positive ion peaks at *m*/*z* 394.13 [M + H]^+^, 416.12 [M + Na]^+^) and ^1^H NMR spectroscopy (the molar ratio of **J_1h_**/**J_1_**). The disappearance of acetal **I_1_** was observed after 2 h of heating. An additional 1 h of heating afforded a mixture of hemiacetal **J_1h_**/aldehyde **J_1_** 1:3.9 with 80.2% conversion of acetal **I_1_**. Prolonged heating over 96 h was accompanied by side reactions leading to only 63.7% of acetal **I_1_** conversion into the desired products in 1:4.8 molar ratio. The formation of two deacetalisation products was confirmed by typical sets of proton signals in the ^1^H NMR spectra originated from the aldehyde and hemiacetal groups, namely sharp singlet of H_22_ at 9.99 ppm in **J_1_** or doublet of O*H* (H_23_) at 6.71 ppm, doublet of H_22_ at 5.38 ppm and singlet of methyl protons H_24_ at 3.33 ppm in molar ratio 1:1:3 in **J_1h_**. Due to the different environments of C_22_ in **J_1h_** and **J_1_** their carbon resonances in the ^13^C NMR spectra are also quite different: 193.78 ppm in **J_1_** and 97.88 ppm in **J_1h_** (Appendix A, ESI †).

#### 3.1.3. Synthesis of 4-((4-(7-((2-Formylpyridin-4-yl)methoxy)-2-oxo-2H-chromene-3-carbonyl)piperazin-1-yl)methyl)picolinaldehyde (**J_2_**) 

Hydrolysis of two acetal groups in *C*_1_ symmetric molecule **I_2_** may theoretically result in eight products: two acetal-hemiacetals, two hemiacetal-aldehydes, two acetal-aldehydes, hemiacetal-hemiacetal and aldehyde-aldehyde **J_2_** (Appendix A, ESI †). ESI MS experiments indicated the formation of a mixture of hemiacetal-hemiacetal (peaks at *m*/*z* 577.26 [M + H]^+^, 599.20 [M + Na]^+^), hemiacetal-aldehyde (peaks at *m*/*z* 545.21 [M + H]^+^, 567.19 [M + Na]^+^) and aldehyde-aldehyde **J_2_** (peaks at *m*/*z* 513.2 [M + H]^+^, 535.19 [M + Na]^+^) from hydrolysis of **I_2_** in water in the presence of 12 M HCl (molar ratio of **I_2_**/HCl 1:6) at 60 °C for 4 h)**.** The collected first fractions upon column chromatography purification (SiO_2_, CH_2_Cl_2_/MeOH 8:1, R_f_ ca. 0.73) showed a mixture of **J_2_** (main product) and a small amount of “hemiacetal and aldehyde” products in relatively good yield (ca. 55%, calculated for **J_2_**), which was used for the condensation reaction with different thiosemicarbazides. The presence of two aldehyde groups in **J_2_** was confirmed by NMR spectroscopy. The two protons gave sharp singlets at 10.01 (H_30_) and 9.99 (H_22_) ppm, whereas two carbon resonances appeared at 193.78 (C_22_) and 193.56 (C_30_) ppm (Appendix A, ESI †).

#### 3.1.4. Synthesis of **HL^1^-HL^3^** and **H_2_L^4^**


The synthesis of TSCs **HL^1^-HL^3^** and **H_2_L^4^** was performed by condensation reactions of the aldehyde **J_1_** with three different thiosemicarbazides in the 1:1 molar ratio in boiling ethanol, while **H_2_L^4^** starting from **J_2_** and 4*N*-dimethylthiosemicarbazide in the 1:2 molar ratio in boiling ethanol, in good yields (65–84%) (Scheme 1 and Appendix A, ESI †). Other attempts described in ESI † and related to Appendix A were less successful. The obtained TSCs **HL^1^-HL^3^** and **H_2_L^4^** were characterised by elemental analysis, multinuclear NMR, UV–vis, IR spectroscopy and ESI mass spectrometry. Mass spectra measured in positive ion mode gave peaks attributed to [M + H]^+^and [M + Na]^+^ions. One- and two-dimensional ^1^H and ^13^C NMR spectra confirmed the expected structures for **HL^1^-HL^3^** and **H_2_L^4^** and the presence of *E*- and/or *Z*-isomers in DMSO-*d*_6_. Whereas in the case of TSCs with the bulky *N*-substituents (R_2_ = Ph in **HL^2^** or R_2_ = 4-hydroxy-3,5-dimethylphenyl in **HL^3^**) only *E*-isomer was observed, for **HL^1^** (R_1_ = R_2_ = Me) both isomers in the molar ratio of *Z*/*E* 1:3.4 are present. These two isomers, shown in [Fig biomolecules-11-00862-ch003] for **HL^1^**, are formed due to hydrogen bond formation between the NH group as proton donor (H_23_ in Appendix A, ESI †), and pyridine nitrogen as proton acceptor, and can be easily distinguished by the resonance of the proton H_23_ in ^1^H NMR spectra. The N*H* signal in *Z*-isomer is downfield shifted in comparison to proton H_23_ of the *E*-isomer. Therefore, the proton H_23_ in *Z*-**HL^1^** and *E*-**HL^1^** resonates at 15.12 and 11.16 ppm, respectively. In the NMR spectra of **HL^2^** and **HL^3^**, only one set of signals is present, and proton H_23_ gave singlets at 12.01 and 11.85 ppm, respectively, which allowed for their assignment to *E*-isomers. The most difficult was the assignment of all resonances for **H_2_L^4^** (Appendix A). In a similar way to **HL^1^** (R_1_ = R_2_ = Me), **H_2_L^4^** gave two sets of signals, originated from the two isomers, namely *Z,Z*- and *E,E*-isomers. Immediately after dissolution in DMSO-*d*_6_, the molar ratio of *Z,Z*/*E,E* was 1:2, but in several hours a new equilibrium was reached, in which the molar ratio between the two isomers was 1:4.3. The N*H* protons involved in the hydrogen bonds (H_31_, H_34_ in Appendix A, ESI †) resonate at 15.12 and 15.06 ppm in *Z,Z***-H_2_L^4^** and at 11.16 and 11.19 ppm in *E,E***-H_2_L^4^**. The presence of *E*- and *Z*-isomers is typical for thiosemicarbazones, and our data are in good agreement with those reported in the literature [24,60].

All obtained coumarin-based TSCs (**HL^1^**–**HL^3^** and **H_2_L^4^)** showed very similar emission spectra in H_2_O, 1% DMSO/H_2_O solutions with a maximum at 451–454 nm (λ_em_) when irradiated with 377–386 nm (λ_ex_) (Appendix A, ESI †). Comparison with the fluorescence spectra of triapine (in water (0.5% DMSO), λ_ex_ = 360 nm, λ_em_ = 457 nm) [58], and 7-hydroxycoumarin-3-carboxylic acid (**G**, λ_ex_ = 350 and 395 nm, λ_em_ = 450 nm, in phosphate buffer (PB), pH 7.4) [59] showed that the combination of two pharmacophores (triapine and coumarin) in the new hybrids **HL^1^**–**HL^3^** and **H_2_L^4^** resulted in a partial quenching of fluorescence, even though the same emission bands with almost the same maxima in the cancer cell-free medium were observed. This fact did not allow for the monitoring of the intracellular distribution of **HL^1^**–**HL^3^** and **H_2_L^4^**.

#### 3.1.5. Synthesis of Complexes

The synthesis of copper(II) complexes was carried out in alcohol (**1**–**3, Cu(HL^1-3^)Cl_2_**) or in DMF (**4, Cu_2_(H_2_L^4^)Cl_4_**) at room temperature by reactions of CuCl_2_·2H_2_O with **HL^1^**-**HL^3^** and **H_2_L^4^** in molar ratio 1:1 (**HL^1^**–**HL^3^**) or 2:1 (**H_2_L^4^**) to give **1**–**4** in good yields (75–82%). Positive ion ESI mass spectra of copper(II) complexes with **HL^1^**-**HL^3^** and **H_2_L^4^** showed peaks at *m*/*z* 556.13, 604.12, 648.16, respectively, attributed to [Cu(L)]^+^or peaks at *m*/*z* 590.12, 638.11, 682.13 in the negative ion mode, respectively, attributed to [Cu(L)Cl–H]^−^. The dicopper(II) complex **4** gave peaks at *m*/*z* 419.02 [Cu_2_(L^4^)]^2+^, 875.06 [Cu_2_(L^4^)Cl]^+^ or *m*/*z* 944.96 [Cu_2_(L^4^)Cl_3_]^−^. Re-crystallisation of **1** from methanol led to crystals of the complex with a monoanionic ligand, namely **Cu(L^1^)Cl·1.775H_2_O** (**1´·1.775H_2_O)**, which was studied by single crystal X-ray diffraction (SC-XRD). Copper(II) forms quite stable complexes with this type of TSC ligands. The log*β*[CuL]^+^ data reported for copper(II) complexes of triapine, pyridine-2-carboxaldehyde 4*N*-dimethylthiosemicarbazone (PTSC) and 3-aminopyridine-2-carboxaldehyde 4*N*-dimethylthiosemicarbazone (APTSC) are equal to 13.89(3), 13.57(2) and 13.95(2) showing that demethylation at the end nitrogen atom of the thiosemicarbazide moiety increases slightly the complex stability [60].

### 3.2. X-ray Crystallography

The result of SC-XRD study of **[Cu(L^1^)Cl]** (**1′**) is shown in Figure 1. The complex crystallised in the triclinic centrosymmetric space group *P*$\bar{1}$ with two crystallographically independent molecules of the complex in the asymmetric unit. The crystal contains co-crystallised water (3.55 molecules per asymmetric unit), which is disorded over several positions. The copper(II) ion adopts a square-planar coordination geometry. The thiosemicarbazone acts as a monoanionic tridentate ligand, which is bound to copper(II) via pyridine nitrogen atom N1, hydrazinic nitrogen atom N2 and thiolato sulfur atom S. The fourth position is occupied by the chlorido co-ligand. The Cu to ligands bond lengths are quoted in the legend to Figure 1. The established coordination mode is typical for copper(II) complexes with pyridine-2-carboxaldehyde thiosemicarbazones [61,62,63]. We refrain from making any comparison of the metrics parameters in this molecule with those in related copper(II) thiosemicarbazonates, taking into account the low-resolution SC-XRD data collected for this compound.

The anticancer activity of the copper(II) complexes of TSCs is often related to their redox reactions with physiological reductants. Therefore, the redox properties of Cu(II) complexes were investigated by cyclic voltammetry (CV) and EPR/UV/Vis-NIR spectroelectrochemistry (SEC) in solution, at room temperature.

### 3.3. Electrochemistry and Spectroelectrochemistry

Cyclic voltammograms (CVs) of **1**–**4** show one irreversible or quasi-reversible one-electron reduction wave between −0.87 and −0.91 V vs. Fc^+^/Fc (Figure 2a and Appendix A in ESI †) that corresponds to the reduction of the Cu(II) to Cu(I) [26,64]. The irreversible oxidation wave at *E*_pa_ = 0.1 V vs. Fc^+^/Fc (Figure 2b) can be only seen for complex **3** with potentially redox-active 2,6-dimethyl-4-aminophenol unit. A similar response was observed for the corresponding proligand **HL^3^** (Appendix A, ESI †), likely indicating oxidation of the 2,6-dimethyl-4-aminophenol moiety [65,66].

In situ spectroelectrochemistry in 0.1 M *n*Bu_4_NPF_6_/DMSO solution provides further evidence for chemical irreversibility of the reduction process, as shown for **2** in Figure 3. UV–vis spectrum of Cu(II) complex **2** exhibits absorption maxima at 338 nm and in the 400–430 nm region. The first maximum can be attributed to intraligand π→π* electronic transition, while other bands at higher wavelengths are due to ligand-to-metal charge-transfer (CT) processes [67]. Absorption spectra measured upon cathodic reduction of **2** in the region of the first reduction peak revealed an increase of the initial optical bands at 338 nm and simultaneous decrease of the of the S→Cu(II) CT band (~420 nm) (Figure 3a). There is only partial recovery of the initial optical band after scan reversal (Figure 3b), which provides further evidence that Cu(I) state is not stable, and the complex decomposes with the ligand (or new ligand product) release [64]. Characteristic for Cu(II), the intensity of the room temperature X-band EPR spectrum of **2** decreased upon stepwise application of a negative potential at the first reduction peak, indicating the formation of a d^10^ EPR-silent Cu(I) species (see inset in Figure 3b). Endogenous thiols are likely able to reduce copper(II) complexes of TSCs [68] to less stable copper(I) species that may dissociate. The released proligand can act as an Fe-chelator as was already unambiguously confirmed for other TSCs, including morpholine–thiosemicarbazone hybrids [26] and 3-amino-2-pyridinecarboxaldehyde-S-methylisothiosemicarbazone [64], and consequently may reduce the tyrosyl radical in R2 RNR. Additionally, thiols may react with Cu(II) ions to form mixtures of Cu(I) and Cu(II) complexes, which could behave either as antioxidants or pro-oxidants, depending on the molar ratio of reactants [69].

### 3.4. Anticancer Activity

The cytotoxic potential of ligands **HL^1^**–**HL^3^** and **H_2_L^4^** and their respective Cu(II) complexes was determined against a range of highly resistant cancer cell lines, including triple negative breast cancer cell line MDA-MB-231, colorectal adenocarcinoma cell line COLO-205 and its multi-drug-resistant analogue COLO-320, as well as healthy human lung fibroblasts MRC-5 by a colorimetric MTT assay, and compared to doxorubicin. The results are presented in Table 2 and Appendix A, ESI †, and represent the mean IC_50_ values with standard deviations. As can be seen from Table 2, in all tested cell lines the ligands and their respective complexes demonstrated antiproliferative activity in a micromolar concentration range, similar to previously published TSC-coumarin hybrids [41]. The cytotoxicity of the ligands showed the following trend in all cancer cell lines: **HL^2^** ≈ **HL^3^ < HL^1^ ≤ H_2_L^4^**, indicating detrimental effects of aromatic substituents in the thioamide group. On the contrary, the addition of the second TSC fragment has improved the cytotoxicity of **HL^1^** by a factor of 1.5–2. The cytotoxicity of the ligand **H_2_L^4^** was comparable to the clinically used doxorubicin in all cancer cells lines, demonstrating its high therapeutic potential. However, this modification has also led to the increased toxicity towards healthy fibroblasts in agreement with high toxicity of 3-AP (selectivity factors (SF) towards MDA-MB-231 and COLO-320 cell lines are 1.3 and 0.9, respectively) [24,25]. In general, **HL^1^–HL^3^** demonstrated higher selectivity towards MRC-5 cells over MDA-MB-231 cells (SF = 3–10). On the contrary, their selectivity towards COLO cell lines was quite poor and only **HL^1^** demonstrated reasonable selectivity reflected by SF = 6.1. The coordination of **HL^1^** and **H_2_L^4^** to Cu(II) centre resulted in a decrease of anticancer activity in all cancer cell lines. Similarly, Cu(II) complexes of 3-AP demonstrated lower cytotoxicity than metal-free 3-AP [70]. However, Cu(II) complexes of **HL^2^** and **HL^3^** retained the cytotoxicity of the ligands in COLO-205 and COLO-320, while being more toxic to MRC-5 (SF = 0.1–0.9). This study demonstrates how slight changes in the 3-AP structure markedly affect the balance between anticancer activity and toxicity to healthy cells.

Since the antiproliferative activity of TSCs is often related to inhibition of the R2 RNR protein, and may involve the formation of iron complexes of TSCs, first the ability of iron(II) complex with **HL^1^** to generate ROS was investigated, followed by the potential of selected compounds prepared in this work to reduce the tyrosyl radical in mR2 RNR.

### 3.5. Redox Activity of **[Fe^II^(L^1^)_2_]**

To investigate whether the iron complex of the lead **HL^1^** hybrid is redox active and therefore able to generate ROS in the aqueous environment, the EPR spin trapping experiments were performed using 5,5-dimethyl-1-pyrrolin-*N*-oxide (DMPO) as the spin trapping agent. The complex **[Fe^II^(L^1^)_2_]** was prepared by the reaction of an anoxic aqueous solution of FeSO_4_·7H_2_O with ethanolic solution of **HL^1^** at 1:2 molar ratio. The formation of **[Fe^II^(L^1^)_2_]** species was confirmed by electronic absorption spectrum of **[Fe^II^(L^1^)_2_]** (Figure 4a), which is characteristic for Fe(II) bis-TSC complexes [24,60] and also by positive ion ESI mass spectrometry of the iron(II) complex under air which revealed the presence of a peak at *m*/*z* 1042 attributed to **[Fe^III^(L^1^)_2_]^+^**.

The EPR spectra of Fe(II)/**HL^1^**/DMPO/H_2_O‒EtOH were recorded either in the presence or absence of H_2_O_2_. In the system containing H_2_O_2_, a four-line EPR signal characteristic for the •OH-DMPO spin adduct was observed (Figure 4b, signal marked with circles). However, also in the absence of H_2_O_2_, the formation of carbon-centred radicals was detected. In this case, both coordinated **HL^1^** and ethanol [71] may act as HO• scavengers, generating carbon-centred radicals which are trapped by DMPO (Figure 4b, signal marked with stars). It is most probable that the coumarin moiety of HL^1^ can readily react with HO• radicals producing carbon-centred radicals. It is important to note that no radicals were formed in the system **HL^1^**/H_2_O_2_/DMPO/H_2_O‒EtOH (4:1, *v*/*v*) (Figure 4b, blue line), which indicates the crucial role of Fe(II) for ROS generation, in both systems (+/− H_2_O_2_). Thus, it may be concluded that complex **[Fe^II^(L^1^)_2_]** is redox active, which may account for the observed antiproliferative activity of **HL^1^** in cancer cell lines.

### 3.6. Tyrosyl Radical Reduction in Mouse R2 RNR Protein

The kinetics of tyrosyl radical reduction in mR2 by the proligand **HL^1^**, and complexes **1** and **3**, were investigated by EPR spectroscopy at 30 K (Figure 5). The results show that **HL^1^** and **1** exhibit similar tyrosyl radical reduction, approximately 25–30% after 10 min of incubation at 1:1 protein-to-compound mole ratio at 298 K. Furthermore, in the presence of an external reductant (DTT) the reduction was increased to 70–75% in 10 min. On the other hand, complex **3** was not as efficient, resulting in only 20% reduction. It is worth noting that the presence of DTT did not have an effect on radical destruction by this complex, indicating that, although the diferric center of mR2 becomes reduced by DTT and therefore causes the tyrosyl radical to be more susceptible for reduction [13,72], complex **3** may be too large to enter the mR2 hydrophobic pocket that harbours the tyrosyl radical, due to the bulky dimethyl-protected phenolic group of the proligand **HL^3^**. Furthermore, when compared to triapine, which reduces 100% tyrosyl radical in 3 min, [64] neither **HL^1^** nor **1** are shown to be efficient mR2 inhibitors. Although **HL^1^** may chelate iron from mR2 and form the redox active **[Fe^II^(L^1^)_2_]** complex, it is evident that this species is devoid of the potent inhibitory activity observed previously for **[Fe^II^(3-AP)_2_****]** [12,13]. This indicates that the introduction of the coumarin/piperazine moiety decreases the radical-reducing potency of 3-AP towards mR2, which may be attributed to the antioxidative properties of the coumarin-moiety of these hybrid compounds.

## 4. Conclusions

An entry to a new series of thiosemicarbazone-coumarin hybrids and their copper(II) complexes via multistep chemical transformations has been provided by this work. The hybrids prepared exhibited antiproliferative activity in human breast adenocarcinoma (MDA-MB-231) and human colorectal adenocarcinoma (COLO-205 and COLO-320) cell lines, with IC_50_ in a micromolar concentration range, in the following rank order: **HL^2^ ≈ HL^3^ < HL^1^ ≤ H_2_L^4^**. These results indicate that aromatic substituents at the thioamide group decrease the antiproliferative activity of the ligand, while the addition of the second TSC fragment (in **H_2_L^4^**) leads to its improvement. However, this addition also led to increased toxicity towards healthy fibroblasts, as previously observed for 3-AP [22,23]. Cu(II) complexes of **HL^1^** and **H_2_L^4^** demonstrated reduced anticancer activities in all cancer cell lines. However, Cu(II) complexes of **HL^2^** and **HL^3^** retained the cytotoxicities of their respective ligands in COLO-205 and COLO-320, while being more toxic to MRC-5. The results from this study confirm once again that the choice of the substituent on 3-AP may greatly affect the balance between anticancer activity and toxicity to healthy cells. It is important to point out that the antiproliferative activity of the lead hybrid **HL^1^** is comparable to that of 3-AP (in MDA-MB-231) and doxorubicin (in COLO-205 and COLO-320), while **HL^1^** is much less toxic to healthy MRC-5 cells than both of these drugs. Regarding mR2 RNR inhibition, the lead hybrid **HL^1^** was shown to be less efficient than 3-AP in tyrosyl radical reduction, which is most likely attributed to the antioxidant properties of the coumarin moiety. It is anticipated that **HL^1^** can chelate iron from mR2, but that the formed iron complex is not a potent inhibitor such as [**Fe^II^(3-AP)_2_****]**. Even though R2 RNR is the biomolecular target for 3-AP, these results indicate that the high antiproliferative activity of **HL^1^** is not due to inhibition of this enzyme. Another likely target is Topoisomerase IIα (Topo IIα), which regulates the DNA topology during cell division [73,74]. α-N-Heterocyclic TSCs were found to bind to the enzyme ATP binding pocket, while their cytotoxicity was found to correlate with Topo IIα inhibition [75]. Moreover, 2-formylpyridine thiosemicarbazones show Topo IIα inhibition activity, which is further increased by complex formation with copper(II) [76]. The key role in this enhancement of the Topo IIa inhibition is the adoption of square-planar coordination geometry, which was also established for compounds investigated in this work.

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
