# Peer review of "Coumarin-Based Triapine Derivatives and Their Copper(II) Complexes: Synthesis, Cytotoxicity and mR2 RNR Inhibition Activity"

_biomolecules, 2021, doi:10.3390/biom11060862_

Round 1
Reviewer 1 Report
see the attached file for comments

Author Response
We are thankful to both reviewers for the issues raised and constructive suggestions.
Reviewer #1:
However, there are some significant concerns with the cytotoxicity assessment, which include:
- The use of normal human lung fibroblasts (MRC-5) is not the best normal cell comparison for the breast cancer (MDA-MD-231) and colorectal adenocarcinoma cell lines. Fibroblasts differ in several regards from normal epithelial cells, making it difficult to assess the degree of selectivity for cancer versus normal cells. Normal breast and colorectal epithelial cells would be a much better direct comparison to the breast and colorectal cancer lines.
Response: We fully agree with the reviewer that MRC-5 is not the best model for normal cells. We have been trying to obtain MCF10A for a long time and unfortunately its import is forbidden due to the required use of cholera toxin. Another option is THLE cells, however, they are extremely expensive and can be used for 2-3 passages only (although the Lonza website states otherwise). To our own disappointment, we cannot solve this problem in a very short time.
- While widely used/misused, there are significant concerns with using the MTT assay as a sole indicator for cytotoxicity. The MTT assay actually measures the activity of a mitochondrial reductase that reduces MTT; it is therefore not a direct indicator of cytotoxicity. To add to this concern, triapine is known to cause selective redox damage to mitochondria in cancer cells, which further complicates interpretation of the MTT results. While it would be ideal to assess if these coumarin analogues cause mitochondrial redox damage, it more be more direct to use a different indicator of cytotoxicity that does not rely on mitochondrial activity. Clonogenic survival is considered a gold standard for assessing the anti-cancer potential of compounds. While more labor-intensive than the colorimetric MTT assay, it is not technically challenging and does not require expensive reagents.
Response: We agree with the reviewer that tetrazolium reduction reflects general metabolic activity or the rate of glycolytic NADH production and not cell viability. If compounds of interest induce mitochondrial damage, the observed MTT results would give false-positive values. However, the results summarized in Table 2 reflect on fairly poor activity of the novel compounds, especially in comparison with previously reported triapine analogues with submicromolar and nanomolar activity; therefore, false-positive results are highly unlikely.
- Their discussion of the results on the cancer selectivity of the coumarin derivatives is rather limited. While HL1 and HL3 have reasonable selectivity for cancer cells (at least based on the MTT assay), HL2 and H2L4 have poor selectivity. These differences could be better clarified.
Response: We have expanded the discussion in agreement with reviewer’s suggestion. We have also added additional column with selectivity factors in the cytotoxicity table.
Anticancer activity. The cytotoxic potential of ligands HL1 -HL3 and H2L4 and their respective Cu(II) complexes was determined against a range of highly resistant cancer cell lines, including triple negative breast cancer cell line MDA-MB-231, colorectal adenocarcinoma cell line COLO205 and its multi-drug resistant analogue COLO-320, as well as healthy human lung fibroblasts MRC-5 by a colorimetric MTT assay, and compared to doxorubicin. The results are presented in Table 2 and Fig. S5, ESI † and represent the mean IC50 values with standard deviations. As can be seen from Table 2, in all tested cell lines the ligands and their respective complexes demonstrated antiproliferative activity in a micromolar concentration range, similar to previously published TSC-coumarin hybrids.72 The cytotoxicity of the ligands showed the following trend in all cancer cell lines: HL2 » HL3 < HL1 ≤ H2L4, indicating detrimental effects of aromatic substituents at the thioamide group. On the contrary, the addition of the second TSC fragment has improved the cytotoxicity of HL1 by a factor of 1.5–2. The cytotoxicity of the ligand H2L4 was comparable to the clinically used doxorubicin in all cancer cells lines, demonstrating its high therapeutic potential. However, this modification has also led to the increased toxicity towards healthy fibroblasts in agreement with high toxicity of 3-AP (selectivity factors (SF) towards MDA-MB-231 and COLO-320 cell lines are 1.3 and 0.9, respectively).22,23 In general, HL1 - HL3 demonstrated higher selectivity towards MRC-5 cells over MDA-MB-231 cells (SF = 3-10). On the contrary, their selectivity towards COLO cell lines was quite poor and only HL1 demonstrated reasonable selectivity reflected by SF = 6.1. The coordination of HL1 and H2L4 to Cu(II) centre resulted in a decrease of anticancer activity in all cancer cell lines. Similarly, Cu(II) complexes of 3-AP demonstrated lower cytotoxicity than metal-free 3-AP.73 However, Cu(II) complexes of HL2 and HL3 retained the cytotoxicity of the ligands in COLO-205 and COLO-320, while being more toxic to MRC-5 (SF = 0.1-0.9). This study has demonstrated how slight changes in the 3-AP structure markedly affected the balance between anticancer activity and toxicity to healthy cells.
Table 2. Cytotoxicity of ligands HL1 -HL3 and H2L4 and corresponding Cu(II) complexes in comparison with doxorubicin and triapine.
|
Compound |
IC50 [µM]a |
SF c |
|||
|
MDA-MB-231 |
COLO-205 |
COLO-320 |
MRC-5 |
|
|
|
HL1 |
4.8 ± 1.3 |
9.2 ± 0.1 |
6.7 ± 0.9 |
41 ± 1 |
6.1 |
|
HL2 |
18 ± 4 |
89 ± 6 |
83 ± 7 |
54 ± 5 |
0.6 |
|
HL3 |
11 ± 1 |
76 ± 8 |
90 ± 10 |
> 100 |
1.1 |
|
HL4 |
3.0 ± 0.3 |
4.4 ± 0.7 |
4.6 ± 0.3 |
4.0 ± 0.3 |
0.9 |
|
Cu(HL1)Cl2 |
25 ± 5 |
12 ± 1 |
14 ± 0 |
11 ± 1 |
0.8 |
|
Cu(HL2)Cl2 |
>100 |
75 ± 7 |
84 ± 8 |
11 ± 1 |
0.1 |
|
Cu(HL3)Cl2 |
67 ± 18 |
58 ± 3 |
59 ± 3 |
53 ± 6 |
0.9 |
|
Cu2(HL4)Cl4 |
14 ± 1 |
11 ± 1 |
9.7 ± 1.1 |
8.7 ± 1.0 |
0.9 |
|
Doxorubicin |
n.ab |
3.3 ± 0.2 |
3.1 ± 0.3 |
5.2 ± 0.2 |
1.7 |
a 50% inhibitory concentrations (IC50) in human breast adenocarcinoma (MDA-MB-231), human colorectal adenocarcinoma (COLO-205 and COLO-320) and human healthy lung fibroblasts (MRC-5), determined by the MTT assay after 72 h exposure. Values are means ± standard deviations (SD) obtained from at least three independent experiments. b n.d. – not detected. c SF is determined as IC50(MRC-5)/IC50(COLO-320).
- Triapine is redox active and is known to generate ROS and perhaps other types of oxidants. Further, there is evidence that Fe(triapine)2 is the cytotoxic species, and not triapine itself. This evidence includes the fact that deferoxamine blocks triapine cytotoxicity (Fe-deferoxamine is not redox active). It would seem relevant to determine if the Fe chelates of their coumarin derivatives are the cytotoxic species.
Response: ROS generation. To investigate whether iron complexes with TSC-coumarin hybrids are able to generate ROS in aqueous environment, the EPR spin trapping experiments were performed by using 5,5-dimethyl-1-pyrrolin-N-oxide (DMPO) as spin trapping agent. [FeII(L1)2] was prepared by the reaction of an anoxic aqueous solution of FeSO4·7H2O with ethanolic solution of HL1 at 1:2 molar ratio. The formation of [FeII(L1)2] species was confirmed by electronic absorption spectrum of [FeII(L1)2] (Fig. 5a), which is characteristic for Fe(II) bis-TSC complexes,24,60 and by positive ion ESI mass spectrometry of the iron(II) complex under air which revealed the presence of a peak at m/z 1042 attributed to [FeIII(L1)2]+.
EPR spectra were recorded either in the presence or in the absence of H2O2. In the system containing H2O2 a four-line EPR signal characteristic for the •DMPO-OH spin adduct (Fig. 5b, signal marked by circles) was observed. Even in the absence of H2O2 we observed the formation of carbon centered spin-adducts. In this case, coordinated ligand and/or ethanol (ethanol was used for solubility reason) acted as a HO• scavengers generating carbon centered radicals which are trapped by DMPO (Fig. 5b, signal marked by stars). We suppose that the coumarin moiety incorporated in the ligand can react with HO• producing new carbon centered radicals. We did not observe any generation DMPO spin adducts in the system HL1/H2O2/DMPO/H2O‒EtOH (4:1, v/v) (blue line in Figure 5b). This fact indicates the role of Fe(II) for ROS generation in both systems. Thus, we may assume that the redox chemistry of [FeII(L1)2] led to the production of different kinds of ROS, which may cause inhibition of R2 RNR protein and/or cancer cell death.
Fig. 5. (a) UV–vis spectrum of [FeII(L1)2] (blue trace) in H2O/EtOH (4:1, v/v) compared to that of HL1 (black trace). (b) Experimental EPR spectra obtained in the system Fe(II)/HL1/DMPO/H2O‒EtOH (4:1, v/v) in the presence of H2O2 (black line) or in the presence of air oxygen (red line), and in the system HL1/H2O2/DMPO/H2O‒EtOH (4:1, v/v) (blue line). Initial concentrations: c(HL1) = 0.2 mM, c(FeSO4·7H2O) = 0.1 mM, c(DMPO) = 0.02 M, c(H2O2) = 0.01 M (circles – •OH-DMPO spin adduct, stars - •R-DMPO spin adducts).
This new data and the highlighted text has been added to the manuscript.
Reviewer 2 Report
The manuscript biomolecules-1227758 "Coumarin-based triapine derivatives and their copper(II) complexes: synthesis, cytotoxicity and mR2 RNR inhibition activity" by Stepanenko et.al. describes the synthesis mono-and dinuclear copper(II) complexes based on a series of thiosemicarbazone-coumarin hybrids and the study of their anti-cancer activity. The structures of all organic compounds and their copper(II) and dicopper(II) complexes were confirmed by modern methods of analysis (elemental analysis, ESI mass spectrometry, 1D and 2D NMR, IR and UV–vis spectroscopies, CV and SEC).
This manuscript describes both a great synthetic work and a large biological part.
Questions and comments:
1) I recommend that the authors strengthen the Introduction part on the design of copper (II) complexes. New articles on the design of copper (II) complexes, as well as their applications, should be added. For example, Pharmaceuticals 2021, 14(3), 244; Inorganics 2021, 9(2), 12; Molecules 2021, 26(8), 2334.
2) Unfortunately, I did not find data on the binding constants of copper (II) complexes This should be added.
Author Response
Reviewer #2:
This manuscript describes both a great synthetic work and a large biological part.
Questions and comments:
1) I recommend that the authors strengthen the Introduction part on the design of copper(II) complexes. New articles on the design of copper (II) complexes, as well as their applications, should be added. For example, Pharmaceuticals 2021,14(3), 244; Inorganics 2021, 9(2), 12; Molecules 2021,26(8), 2334.
Response: Several sentences were added and the recommended articles have been cited: “Synthetic nucleoside analogues and Schiff bases are used as suitable models for investigation of nucleic acids and as chelating agents for application in different fields of research. Schiff bases are easily prepared and their electronic and steric properties can be fine-tuned for biomedical applications.”
“Copper(II) complexes with potentially tetradentate piperazine ligands bearing pendant pyridyl groups were reported to effectively cleave DNA and to be cytotoxic.”
2) Unfortunately, I did not find data on the binding constants of copper(II) complexes This should be added.
Response: Copper(II) forms quite stable complexes with this type of TSC ligands. The logb [CuL]+ data reported for copper(II) complexes of triapine, pyridine-2-carboxaldehyde 4N-dimethylthiosemicarbazone (PTSC) and 3-aminopyridine-2-carboxaldehyde 4N-dimethylthiosemicarbazone (APTSC) are equal to 13.89(3), 13.57(2) and 13.95(2) showing that demethylation at the end nitrogen atom of the thiosemicarbazide moiety increases slightly the complex stability [EJIC 2010, 1717-1728].
Round 2
Reviewer 2 Report
I thank the authors for answering my questions and improving the manuscript.